# Calcium Requirement of *Yunnan Semi-fine Wool* Rams (*Ovis aries*) Based on Growth Performance, Calcium Utilization, and Selected Serum Biochemical Indexes

**DOI:** 10.3390/ani14111681

**Published:** 2024-06-05

**Authors:** Xiaojun Ni, Xiaoqi Zhao, Baiji Danzeng, Yinjiang Li, Allai Larbi, Hongyuan Yang, Yuanchong Zhao, Zhengrong You, Bai Xue, Guobo Quan

**Affiliations:** 1Yunnan Animal Science and Veterinary Institute, Kunming 650224, China; nixiaojun123@126.com (X.N.); zhaoxiaoqi2023@163.com (X.Z.); danzeng1376@163.com (B.D.); li2893@sina.com (Y.L.); maomao7836@sohu.com (H.Y.); 2Yunnan Provincial Engineering Research Center of Animal Genetic Resource Conservation and Germplasm Enhancement, Kunming 650224, China; 3Laboratory of Sustainable Agriculture Management, Higher School of Technology Sidi Bennour, Chouaib Doukkali University, El Jadida 24000, Morocco; allay.larbi@gmail.com; 4Qiaojia Agriculture and Rural Bureau, Qiaojia 654699, China; qjzych@126.com; 5Zhaotong Agriculture and Rural Bureau, Zhaotong 657099, China; youzhr1968@163.com; 6Institute of Animal Nutrition, Sichuan Agricultural University, Chengdu 611130, China

**Keywords:** average daily gain, calcium retention, fecal calcium, nutritional requirement, serum indexes, sheep

## Abstract

**Simple Summary:**

*Yunnan Semi-fine wool sheep* is a recently developed breed of sheep in China for meat and wool production. However, the nutrient requirements of this breed are scarcely studied, especially regarding minerals. This study aims to evaluate the effects of dietary calcium levels on the growth performance, calcium utilization, and serum biochemical indexes of *Yunnan semi-fine wool* rams as well as to estimate their calcium requirement. Diets with calcium levels ranging from 0.73% to 0.89% can improve the growth performance and calcium utilization efficiency of rams, which could serve as baseline when formulating diets for *Yunnan semi-fine wool* rams.

**Abstract:**

Calcium (Ca) is required for the growth and development of sheep, but the requirement of *Yunnan semi-fine wool* (YSW) rams remains uncovered. The current study aims to estimate the Ca requirement of growing YSW rams based on their growth performance, Ca utilization, and serum biochemical indexes. Forty-five YSW rams (10-month-olds) were randomly allocated to five dietary treatments with varying Ca levels of 0.50% (D1), 0.68% (D2), 0.73% (D3), 0.89% (D4), and 0.98% (D5). A higher value for average daily gain and a lower value for the feed conversion ratio were observed in the D3 group compared to the D5 group (*p* < 0.05). The dry matter intake amount changed quadratically with the increased Ca levels (*p* < 0.05). The levels of Ca intake, fecal Ca, and excreted Ca were significantly higher in the D5 group than those in the D1 group (*p* < 0.05). The apparent Ca digestibility rate and the Ca retention rate were significantly higher in the D4 group than in the D1 group (*p* < 0.05). The serum Ca concentration increased linearly with the incremental levels of dietary Ca (*p* < 0.05). The activity of alkaline phosphatase was significantly higher in the D1 group than in the D2 group (*p* < 0.05). The serum levels of hydroxyproline, osteocalcin, and calcitonin decreased from the D1 group to the D2 group, and then significantly ascended (*p* < 0.05) with the dietary Ca levels from the D3 group to the D5 group. The serum parathyroid hormone content was elevated from the D1 group to the D3 group and then decreased from the D4 group to the D5 group. After calculation, the daily net Ca requirement for the maintenance of YSW rams was 0.073 g/kg of BW^0.75^, and the daily total Ca requirement was 0.676 g/kg of BW^0.75^. To optimize the growth performance and the Ca utilization of YSW rams, the recommended dietary Ca level ranges from 0.73% to 0.89% based on this study.

## 1. Introduction

*Yunnan semi-fine wool sheep* (YSW), a famous Chinese dual-purpose sheep breed used for the production of meat and wool [1], is characterized by strong adaptability, a fast growth rate, and strong disease resistance, and it is therefore commonly raised in high-altitude and cold mountainous areas [2,3]. However, the nutrient requirements of this breed are scarcely studied, especially regarding minerals. Calcium (Ca) is one of the critical minerals necessarily required for ruminant growth and bone development [4]. Dietary Ca deficiency can lead to disruptions in the cellular process and bone formation, metabolic disorders, and ultimately poor performance in the animals [5]. So far, it can be seen that less information regarding Ca requirements can result in lower feed efficiency in YSW sheep. Therefore, suitable dietary Ca levels redound to Ca digestion and absorption, thus improving the growth performance of sheep [6,7].

According to ARC (1980) [8], the net Ca requirement for the maintenance of sheep was 0.016 g/kg of body weight (BW), and the amount of Ca retained in the weight gain of lambs was 11.00 g/kg of empty body weight gain (EBWG). The NRC (2007) [9] then used the equation developed by AFRC (1991) to estimate the Ca requirements of sheep. But the Ca requirement differs greatly among sheep breeds. Pereira et al. [10] reported that the net Ca requirement for the maintenance of Somali lambs is 0.030 g/kg of BW, which is nearly twice the value reported by ARC (1980). Herbster et al. [11] suggested that the net Ca requirement for the maintenance of male hair sheep is 0.024 g/kg of BW, which is 50% greater than the value reported by ARC (1980). Additionally, some studies have demonstrated that the net Ca requirements to achieve weight gain in growing German merino lambs and Dorper crossbred sheep are 14.00 g/kg of EBWG [12] and 10.52~12.35 g/kg of EBWG [13], respectively. Therefore, it can be concluded from these reports that the Ca requirements of sheep vary greatly among breeds and physiological stages. In addition, a recent review summarized the research progress on the nutrient requirements of sheep in China, indicating that more investigations are needed to obtain the optimal mineral requirements [14]. Thus, it is necessary to study the Ca requirements of YSW sheep for optimal performance. 

In recent years, our research team has determined the Ca requirements of YSW sheep during pregnancy and lactation [15,16]; however, the requirements of YSW rams during the growth stage are still being investigated. The current study aimed to evaluate the effects of dietary Ca levels on the growth performance, Ca utilization, serum biochemical indexes, and Ca requirement of YSW rams. The outcomes of this study will be helpful in guiding the efficient feeding of YSW sheep. 

## 2. Materials and Methods

### 2.1. Animals, Diets, and Experimental Design

The experiment was conducted at the farm of Kunming Yixingheng Livestock Technology Co., Ltd. (Kunming, China). All experimental animals and procedures were conducted in accordance with the Chinese Guidelines for Animal Welfare and were approved by the Ethics Committee of the Yunnan Animal Science and Veterinary Institute (202106002). Moreover, all authors strictly followed the approved protocols and guidelines by the State Science and Technology Commission of the People’s Republic of China, 1988, and the Standing Committee of Yunnan Provincial People’s Congress 2007.

Forty-five 10-month-old YSW rams with an average body weight of 40.37 ± 0.49 kg were randomly assigned to 5 groups, with 9 replicates per group. The 5 groups of rams were subject to 5 dietary treatments with varying Ca levels of 0.50% (D1), 0.68% (D2), 0.73% (D3), 0.89% (D4), and 0.98% (D5), respectively. The D3 group was fed an optimum dietary Ca level that was described by the NRC (2007) [9] to meet the nutritional requirements of sheep with a body weight of 40 kg and a daily gain of 250 g/d, which served as the control, while groups D1 and D2 were fed lower levels, and groups D4 and D5 were fed upper levels that were tested to estimate the Ca requirements of YSW rams. The experimental diets, as shown in Table 1, were formulated according to the NRC (2007). The experiment lasted for 44 d, including a 14 d dietary transition period followed by a 30 d feeding trial.

### 2.2. Feeding Management

The sheep house was cleaned and disinfected before the start of the feeding trial. A total of 45 experimental rams were housed indoors in individual pens. The rams were weighed and grouped before the start of the preliminary period, and they were fed twice daily at 08:00 and 17:00 h. All rams were fed ad libitum and provided adequate fresh water. The daily feeding amount of the rams was adjusted according to the feed intake of the previous day and ensured a 10% refusal. The rams were weighed at the beginning and end of the formal period to obtain the initial body weight (IBW) and the final body weight (FBW), and the average daily gain (ADG) of the rams was calculated. The dry matter intake (DMI) of each group was calculated according to the feed supplied and refused. The feed conversion ratio (FCR) was calculated as the ratio of DMI to ADG.

### 2.3. Digestion and Metabolism Experiment

The digestion and metabolism trial was conducted from the 11th to the 15th day of the formal period, during which 5 rams from each group were randomly selected and kept in a single metabolic cage (1.8 m × 0.8 m) for the total collection of feces and urine. A total of 20% of the fecal samples from each sheep was collected, and the fecal samples collected for the determination of the crude protein contents were added with 10% dilute hydrochloric acid (HCl) for nitrogen fixation. Collected urine samples were filtered, weighed, and recorded with a measuring cylinder, and then 10% of the sample was mixed with 10% HCl. At the end of the trial period, 10 mL of blood samples from all rams was collected from the jugular venipuncture to conduct a serum biochemical analysis. All collected feces, urine, and blood samples were stored at −20 °C for later analysis. 

### 2.4. Chemical Analysis

The contents of Ca and P in the diet, feces, and urine were analyzed according to the national standards of GB/T 6436-2018 [17] (Ca) and GB/T 6437-2018 [18] (P). Blood samples were centrifuged at 3000× *g* for 15 min at 4 °C, and the supernatant was collected and stored at −80 °C until usage to determine serum biochemical indexes. The activity of alkaline phosphatase (ALP), and the contents of Ca, P, hydroxyproline (HYP), osteocalcin (OC), calcitonin (CT), and parathyroid hormone (PTH) in the serum of rams were detected using enzyme-linked immunosorbent assay (ELISA) kits (Beijing SINO-UK Institute of Biological Technology, Beijing, China) according to the manufacturer’s instructions. 

### 2.5. Statistical Analysis and Calculation

All data are presented as the mean values and standard error of the mean (SEM). A statistical analysis was performed using the SPSS 22.0 software. One-way analysis of variance (ANOVA) was used to analyze the data related to growth performance, Ca utilization, and serum biochemical indexes. Tukey’s HSD test was used for the comparison of the means. The linear and quadratic effects of the treatments were obtained following an analysis of variance. The linear regression analyses were used to estimate the net Ca requirements for the maintenance of sheep. A P value of less than 0.05 was considered statistically significant.

The main indicators were calculated as follows: Digestible Ca (g/d) = Ca intake − fecal Ca;Ca apparent digestibility (%) = (digestible Ca/Ca intake) × 100;Excreted Ca (g/d) = fecal Ca + urinary Ca;Retained Ca (g/d) = Ca intake − fecal Ca − urinary Ca;Ca retention rate (%) = (retained Ca/Ca intake) × 100.

## 3. Results

### 3.1. Growth Performance

The growth performance of the rams is presented in Table 2. There were no significant differences in the IBW and FBW among the five groups (*p* > 0.05). The ADG was greater and the FCR was lower in the D3 group than in the D5 group (*p* < 0.05). The ADG and FCR among the D1, D2, and D4 groups had no significant differences (*p* > 0.05). The DMI changed quadratically with the increased Ca levels (*p* < 0.05), and the DMI in the D2 group was the highest, which was significantly higher than the D1 group (*p* < 0.05). 

### 3.2. Ca Intake, Excretion, and Digestibility

The results of Ca intake, excretion, and digestibility are presented in Table 3. The levels of Ca intake, fecal Ca, and excreted Ca increased linearly with the increased dietary Ca concentrations (*p* < 0.05). The values of fecal and excreted Ca in the D5 group were significantly higher than those in the D1 group (*p* < 0.05). No differences were found in the urinary Ca levels among all groups (*p* > 0.05). The digestible Ca, retained Ca, apparent Ca digestibility, and Ca retention rates showed linear and quadratic changes with the increase in Ca levels (*p* < 0.05). In addition, the results showed that the D4 group had the highest value of apparent Ca digestibility and the highest Ca retention rate, and they were significantly higher than those the D1 group (*p* < 0.05). 

### 3.3. Serum Biochemical Indexes

The effects of the dietary Ca levels on the serum biochemical indexes are shown in Table 4. The results indicate that the serum Ca concentration increased linearly with the incremental levels of dietary Ca (*p* < 0.05). There was no significant difference in the serum P concentration among the five groups (*p* > 0.05). The activity of the ALP content in the D2 group was significantly lower than that in the D1 group (*p* < 0.05). The serum levels of HYP, OC, CT, and PTH were quadratically altered with the dietary Ca levels (*p* < 0.05). The serum HYP, OC, and CT concentrations decreased from the D1 group to the D2 group and then significantly ascended with the dietary Ca levels from the D3 group to the D5 group (*p* < 0.05), and the serum PTH concentration increased from the D1 group to the D3 group and then decreased from the D4 group to the D5 group (*p* < 0.05). 

### 3.4. Dietary Ca Requirements

The relationships between the daily Ca intake (g/kg of BW^0.75^) and the fecal Ca (g/kg of BW^0.75^), the excreted Ca (g/kg of BW^0.75^), and the retained Ca (g/kg of BW^0.75^) of the YSW rams are shown in Figure 1, Figure 2, and Figure 3, respectively. It was found that the fecal Ca, the excreted Ca, and the retained Ca were linearly correlated with the Ca intake. The daily net Ca requirement for maintenance was estimated from the intercept of the linear regression between the amount of retained Ca and the Ca intake, which was 0.073 g/kg of BW^0.75^, and the retention coefficient was 0.388. The total Ca requirement of the rams was estimated based on the ADG of the groups with five dietary Ca levels. The maximum ADG was obtained when the dietary Ca level was 0.73%. Therefore, the daily total dietary Ca requirement of the YSW rams was determined as 0.676 g/kg of BW^0.75^.

## 4. Discussion

In this study, the DMI of the YSW rams increased when the dietary Ca levels ranged from 0.50% to 0.68%, but it decreased when the dietary Ca levels rose to 0.98%. Similarly, in a previous study, the DMI of growing rams increased when the dietary Ca levels increased from 0.33% to 0.71% but decreased when the dietary Ca levels increased to 1.09% [19]. In this study, the ADG of the YSW rams was the highest when they were fed a diet with a 0.73% Ca level, and then it significantly decreased when the dietary Ca level was increased to 0.98%. In another study, the same trend was found in the ADG of growing rams when they were fed dietary Ca levels ranging from 0.13% to 0.97% [20]. Several studies established that a higher level of Ca feeding suppressed the feed intake [21], caused urinary calculi [22], and led to poor performance [23]. So, these studies suggest that excessive dietary Ca levels (as D5 group in current experiment) may reduce feed intake and affect the growth performance of rams. In addition, our research found that the FCR value of the rams was the lowest when the dietary Ca level was 0.73%, indicating a high feed conversion ratio of sheep at this concentration, while the FCR value was increased when the dietary Ca level was elevated to 0.98%. However, some previous studies have reported that increasing the dietary Ca and P levels at the same time has no significant effects on the DMI, ADG, and FCR of growing lambs [24,25], indicating that it may be related to the dietary Ca and P balance. It has been demonstrated that the digestion and metabolism of Ca were affected by the dietary Ca levels [26,27]. In the current study, with the increase in the Ca contents in the diets, the fecal Ca and excreted Ca of the YSW rams increased linearly, and the digestible Ca, retained Ca, apparent Ca digestibility, and the retention rate changed quadratically. Previous studies demonstrated that when increasing the dietary Ca contents, the fecal Ca, the absorbed Ca, and the retained Ca of castrated male goats were linearly increased [21], the excreted Ca of Chahar lambs (German Merino × Inner Mongolia fine wool sheep) were significantly increased [28], and the apparent Ca digestibility of growing goats showed a quadratic change [20]. The difference in trends in the Ca digestibility and the retention rate can be attributed to the different breeds used, physiological stage, and Ca requirement. Restricting or oversupplying dietary Ca may alter its bioavailability in sheep [9]. Our results show that the apparent Ca digestibility and the retention rate decreased when the used dietary Ca levels increased from 0.89% to 0.98%, indicating that a high Ca diet can reduce Ca bioavailability, while a 0.89% dietary Ca level can improve its utilization by YSW rams.

The serum Ca, P, PTH, HYP, OC, and CT concentrations and ALP activity were important parameters to evaluate Ca metabolism, the nutritional status, and requirements in ruminants [29]. Several studies have pointed out that the serum Ca and P contents in sheep generally range from 2.25 to 3.00 mmol/L and 1.29 to 2.90 mmol/L [5,30]. In this study, when the dietary Ca levels were in the range between 0.50% and 0.98%, the serum Ca content of the rams changed from 2.61 to 2.89 mmol/L, and the serum P content changed from 1.53 to 2.04 mmol/L, which are all within the above reference ranges. It was found that the serum Ca content increased with the increased dietary Ca levels, while the serum P content was not significantly changed. Several previous studies also reported the same results [24,31], indicating that the serum Ca content of sheep was susceptible to changes in the dietary Ca levels. The hormones of PTH, CT, and 1,25-dihydroxycholecalciferol are involved in the regulation of Ca metabolism in the blood and have regulatory roles in its absorption, retention, and excretion and other metabolic processes [5]. In this study, the serum PTH concentration increased with dietary Ca levels ranging from 0.50% to 0.73% and then decreased. This may have been due to the increase in the serum Ca content of sheep, leading to the reduction in the serum PTH content to regulate the balance of the blood Ca level. The serum concentrations of CT, OC, and HYP decreased with dietary Ca levels ranging from 0.50% to 0.68% and then ascended with dietary Ca levels up to 0.98%. The OC, HYP, and ALP concentrations can reflect the Ca metabolism of animals [32,33]. Previously, a study showed that the serum OC and ALP contents of growing sheep and goats increased with the increase in dietary Ca concentrations [34]. In addition, the bone renewal rate of growing YSW rams is fast, which may lead to the increase in the serum OC and HYP contents. In addition, it was found that the serum ALP activity was negatively correlated with the change in the serum P content. 

In this study, the daily net Ca requirement for the maintenance of YSW rams is 0.073 g/kg of BW^0.75^, which is equivalent to 0.028 g/kg of BW. Based on the total endogenous loss of Ca, the ARC (1980) estimated that the daily net Ca requirement for the maintenance of sheep is 0.016 g/kg of BW [8]. Recent research carried out by Herbster et al. [11] suggests that the daily net Ca requirement for the maintenance of wool sheep is 0.024 g/kg of BW, which is 50% greater than that of the ARC (1980) value but slightly lower than the results in our study. In this study, the net Ca requirement for maintenance is similar to that reported by Pereira et al. for Somali lambs [10], and it is lower than that reported by Sousa et al. for Santa Ines sheep, which is 0.036 g/kg of BW [27]. In addition, Vargas et al. [35] pointed out that the daily net Ca requirement for the maintenance of goats, estimated by the minimum endogenous losses method (0.038 g/kg of BW), is greater than that estimated using the comparative slaughter technique (0.021 g/kg of BW). Therefore, the differences in the net Ca requirements for maintenance in these studies may be due to variations in the breeds, growth stages, and estimation methods used. 

The maintenance requirements of dietary Ca were affected by the efficiency of nutrient absorption and utilization in feed. When the Ca retention coefficient was considered, the daily dietary Ca requirement for the maintenance of YSW rams was 0.187 g/kg of BW^0.75^, which is equivalent to 0.073 g/kg of BW. According to the sheep nutrient requirements in the study by the NRC (2007) [9], the formula for calculating the Ca requirement for the maintenance of sheep is Cam, g/d = (0.623*DMI + 0.228)/0.40. The DMI of YSW rams in this experiment was 1.30~1.55 kg/d, and the daily Ca requirement for maintenance was calculated to be 0.059~0.068 g/kg of BW, which is lower than the results in this study. A greater daily Ca requirement for the maintenance of lambs was found, namely 0.138 g/kg of BW [10], which is almost twice the value calculated by the NRC (2007). The daily Ca requirement for the maintenance of hair sheep was calculated to be 0.055 g/kg of BW [27], which is lower than the results in our study. This may be caused by the different Ca utilization efficiency of sheep breeds at different physiological stages.

In this study, the ADG was used as a marker to estimate the total Ca requirement of YSW rams by using a gradient feeding trial. The daily total Ca requirement was estimated to be 0.676 g/kg of BW^0.75^ based on the group with the highest ADG (278.43 g/d), which resulted in the total Ca requirement of 10.75~12.71 g/d for 40-50 kg YSW rams. The NRC (2007) recommends a Ca requirement of 4.80–4.90 g/d for 8-month-old growing rams of 40–50 kg with a daily gain of 250 g [9], and Herbster et al. [11] suggested a total Ca requirement of 4.70~5.16 g/d for 30 kg sheep gaining 200 g/d. The total Ca requirement value calculated in this study is almost twice the values calculated by the NRC (2007) and Herbster et al. In addition, Ma et al. reported that the Ca requirement of 35~50 kg Dorset × thin-tailed Han rams is 10.70–11.60 g/d [14], which is close to our results. According to newly released nutrient requirements of mutton sheep and goat (NY/T 816-2021) in China, the Ca requirements for 40~50 kg rams with ADGs of 200 g/d and 300 g/d are 12.70~14.50 g/d and 13.30~14.90 g/d, respectively [36], which are higher than our results.

## 5. Conclusions

Based on the results, feeding 0.73% dietary Ca improved the feed intake and growth performance in terms of the average daily gain of YSW rams. Moreover, feeding 0.89% improved Ca digestibility and its retention. The daily net Ca requirement for maintenance was 0.073 g/kg of BW^0.75^, and the daily total Ca requirement was 0.676 g/kg of BW^0.75^. These insights can assist in formulating nutritionally balanced diets that more effectively meet the Ca requirements of YSW sheep, potentially optimizing their overall health and productivity. The current experiment had limitations in measuring the influence of Ca on muscle and in conducting a bone Ca analysis along with rumen microbiota. Therefore, further studies are warranted to evaluate how dietary Ca levels can influence body composition and rumen health.

## Figures and Tables

**Figure 1 animals-14-01681-f001:**
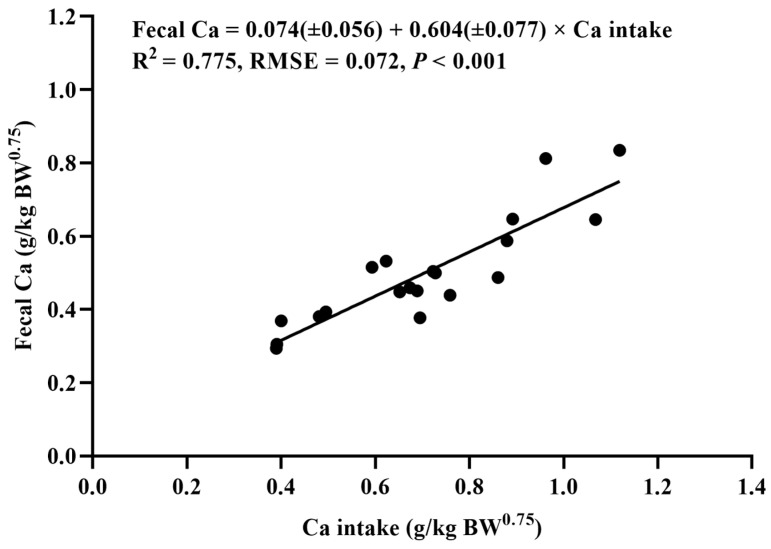
The relationship between the daily Ca intake (g/kg of BW^0.75^) and the daily fecal Ca (g/kg of BW^0.75^) of *Yunnan semi-fine wool* rams. RMSE: root mean square of error.

**Figure 2 animals-14-01681-f002:**
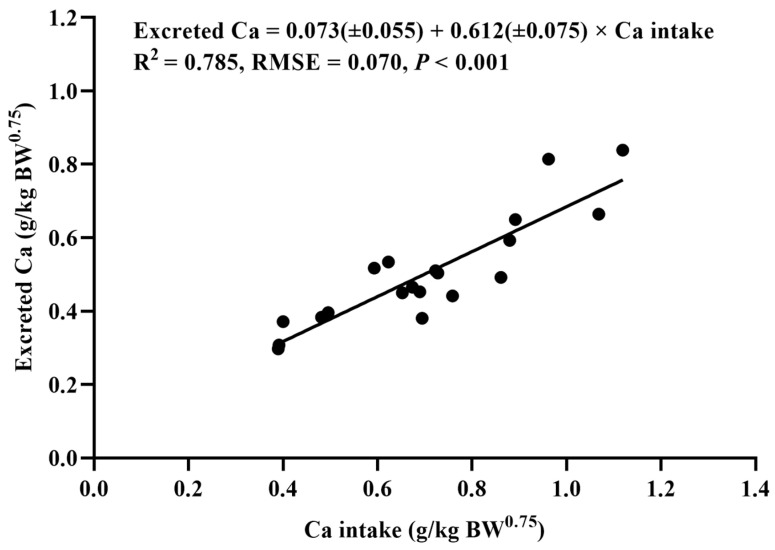
The relationship *between* the daily Ca intake (g/kg of BW^0.75^) and the daily excreted Ca (g/kg of BW^0.75^) of *Yunnan semi-fine wool* rams. RMSE: root mean square of error.

**Figure 3 animals-14-01681-f003:**
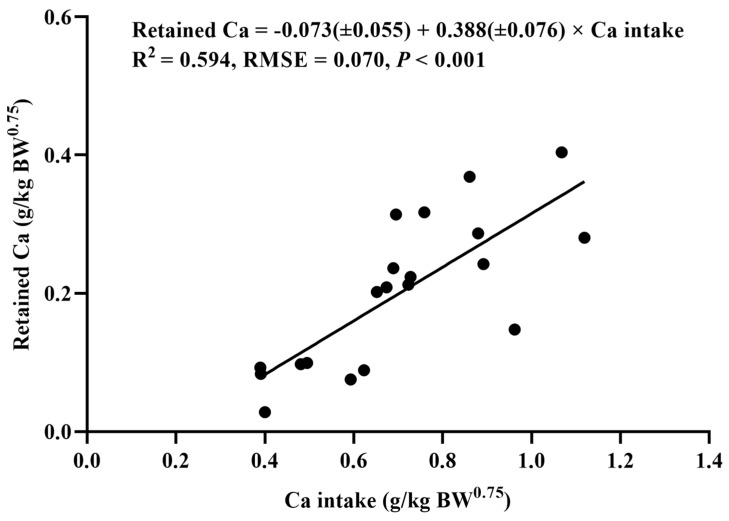
The relationship between the daily Ca intake (g/kg of BW^0.75^) and the daily retained Ca (g/kg of BW^0.75^) of *Yunnan semi-fine wool* rams. RMSE: root mean square of error.

**Table 1 animals-14-01681-t001:** Compositions and nutrient levels of experimental diets.

Items	Experimental Diets				
	D1	D2	D3	D4	D5
Ingredients (%)					
Corn	28.00	13.00	15.00	12.70	13.50
Wheat bran	0.00	3.40	2.50	7.50	6.95
Soybean meal	9.15	13.80	10.00	8.95	10.20
Rapeseed meal	0.00	0.00	4.30	1.40	1.60
Corn starch	12.60	18.00	16.60	16.00	14.90
CaCO_3_	0.00	0.20	0.50	0.95	1.35
CaHPO_4_	0.00	0.10	0.10	0.00	0.00
NaH_2_PO_4_	0.15	0.00	0.00	0.00	0.00
NaCl	0.20	0.50	0.50	0.50	0.50
NaHCO_3_	0.30	0.00	0.00	0.00	0.00
Premix ^1^	1.00	1.00	1.00	1.00	1.00
Corn silage	12.60	27.00	27.00	26.00	30.00
Broad bean stalk	16.00	0.00	0.00	9.00	2.00
Wheat straw	20.00	23.00	22.50	16.00	18.00
Total	100.00	100.00	100.00	100.00	100.00
Nutrient levels ^2^					
ME (MJ/kg)	9.31	9.34	9.30	9.31	9.30
CP (%)	10.70	10.16	11.02	10.59	11.01
NDF (%)	33.49	33.00	35.65	34.94	35.18
ADF (%)	23.07	19.79	22.12	21.00	21.00
Ca (%)	0.50	0.68	0.73	0.89	0.98
P (%)	0.30	0.29	0.26	0.26	0.26

^1^ The premix provided the following per kg of diets: Mn, 58 mg; Fe, 145 mg; Zn, 80 mg; Cu, 10 mg; I, 2.5 mg; Se, 0.35 mg; Co, 0.65 mg; VA, 10,000 IU; VD3, 1000 IU; and VE, 50 IU. ^2^ ME was a calculated value, while the others were measured values. ME: metabolic energy, CP: crude protein, NDF: neutral detergent fiber, ADF: acid detergent fiber, Ca: calcium, P: phosphorus.

**Table 2 animals-14-01681-t002:** The effects of the dietary Ca levels on the growth performance of *Yunnan semi-fine wool* rams.

Items	Experimental Diets					SEM	*p*-Value		
	D1	D2	D3	D4	D5		Treatment	Linear	Quadratic
IBW (kg)	40.13	40.95	40.49	40.83	39.53	0.460	0.874	0.657	0.611
FBW (kg)	46.76	47.51	48.84	47.39	45.50	0.544	0.430	0.460	0.170
ADG (g/d)	221.09 ^ab^	218.67 ^ab^	278.43 ^a^	218.83 ^ab^	199.21 ^b^	8.153	0.026	0.420	0.068
DMI (kg/d)	1.30 ^b^	1.55 ^a^	1.52 ^a^	1.50 ^a^	1.49 ^a^	0.023	0.004	0.072	0.005
FCR	6.03 ^ab^	7.46 ^ab^	5.49 ^b^	7.15 ^ab^	7.75 ^a^	0.252	0.013	0.084	0.156

Values in the same row with different small letter superscripts are significantly different (*p* < 0.05). SEM: standard error of the mean (overall). IBW: initial body weight; FBW: final body weight; ADG: average daily gain; DMI: dry matter intake; FCR: feed conversion ratio.

**Table 3 animals-14-01681-t003:** The effects of dietary Ca levels on the Ca intake, excretion, and digestibility of *Yunnan semi-fine wool* rams.

Items	Experimental Diets					SEM	*p*-Value		
	D1	D2	D3	D4	D5		Treatment	Linear	Quadratic
Ca intake (g/d)	7.54 ^c^	11.45 ^b^	12.23 ^b^	15.47 ^a^	16.50 ^a^	0.775	<0.001	<0.001	<0.001
Fecal Ca (g/d)	5.88 ^b^	8.22 ^ab^	7.62 ^ab^	8.36 ^ab^	10.18 ^a^	0.472	0.037	0.003	0.015
Urinary Ca (g/d)	0.05	0.05	0.08	0.08	0.05	0.005	0.151	0.658	0.189
Excreted Ca (g/d)	5.94 ^b^	8.27 ^ab^	7.70 ^ab^	8.44 ^ab^	10.28 ^a^	0.477	0.036	0.003	0.015
Digestible Ca (g/d)	1.65 ^c^	3.23 ^bc^	4.62 ^abc^	7.11 ^a^	6.32 ^ab^	0.526	<0.001	<0.001	<0.001
Retained Ca (g/d)	1.60 ^c^	3.18 ^bc^	4.54 ^abc^	7.03 ^a^	6.22 ^ab^	0.524	<0.001	<0.001	<0.001
Apparent Ca digestibility (%)	22.05 ^b^	27.96 ^ab^	37.49 ^ab^	45.50 ^a^	38.97 ^ab^	2.583	0.019	0.003	0.004
Ca retention rate (%)	21.35 ^b^	27.55 ^ab^	36.83 ^ab^	45.01 ^a^	38.38 ^ab^	2.599	0.019	0.003	0.004

Values in the same row with different small letter superscripts are significantly different (*p* < 0.05). SEM: standard error of the mean (overall).

**Table 4 animals-14-01681-t004:** The effects of the *dietary* Ca levels on the serum biochemical indexes of *Yunnan semi-fine wool* rams.

Items	Experimental Diets					SEM	*p*-Value		
	D1	D2	D3	D4	D5		Treatment	Linear	Quadratic
Ca (mmol/L)	2.64 ^ab^	2.61 ^b^	2.73 ^ab^	2.80 ^ab^	2.89 ^a^	0.032	0.025	0.011	0.041
P (mmol/L)	1.77	1.73	1.53	1.99	2.04	0.065	0.058	0.175	0.081
ALP (U/L)	339.45 ^a^	232.80 ^b^	327.74 ^ab^	289.42 ^ab^	330.06 ^ab^	13.121	0.020	0.629	0.541
HYP (µg/mL)	12.20 ^c^	9.94 ^e^	11.12 ^d^	13.38 ^b^	14.53 ^a^	0.340	<0.001	0.002	<0.001
OC (ng/mL)	5.43 ^b^	3.83 ^c^	4.56 ^c^	5.86 ^b^	6.69 ^a^	0.216	<0.001	0.014	<0.001
CT (ng/L)	86.69 ^bc^	68.66 ^d^	81.93 ^c^	98.06 ^b^	110.85 ^a^	3.131	<0.001	0.004	<0.001
PTH (pg/mL)	110.06 ^b^	170.01 ^a^	171.90 ^a^	91.47 ^b^	86.16 ^b^	8.532	<0.001	0.296	0.005

Values in the same row with different small letter superscripts are significantly different (*p* < 0.05). SEM: standard error of the mean (overall). Ca: calcium; P: phosphorus; ALP: alkaline phosphatase; HYP: hydroxyproline; OC: osteocalcin; CT: calcitonin; PTH: parathyroid hormone.

## Data Availability

The data supporting the findings of this study are available upon request from the corresponding author.

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
