# Peer review of "Calcium Requirement of *Yunnan Semi-fine Wool* Rams (*Ovis aries*) Based on Growth Performance, Calcium Utilization, and Selected Serum Biochemical Indexes"

_animals, 2024, doi:10.3390/ani14111681_

Round 1
Reviewer 1 Report
Comments and Suggestions for Authors
There are a few points that need to be addressed first.
L18: Shouldn't have a final point after "China"?
L22: Shouldn't be a comma or start "which" with capital?
Keywords: Should be in alphabetical order.
L50: What "Ca" mean? You must describe an acronym before it use. The acronym used in the abstract don't count.
L88: This citation form is correct?
Introduction: Ok, the calcium requirements vary throughout different animal breeds. However, what problems can this variability cause? I suggest you include some words describing how the provision of unbalanced calcium can disturb animals health. With emphasis to the fact that using either a greater or lesser amount of requirements could lead to problems.
L91: You describe the use of NRC to estimate the initial animals requirements, however, in the introduction you didn't discussed the NRC, only ARC. The point here is the fact that to us, readers, we don't know if the NRC has a greater or smaller Ca requirements than ARC. Other point is the ARC is from 1988, while NRC is from 2007. This already don't allow a better data by itself?
L92: Only 30 days? This is enough to a requeirements experment? Other point is the fact that in your title you state that will determinat the Ca requeirements troughout the animals' growth, however you used the animals with initial BW of 40kg.
L86-92: Which was the control level? The level used only to mantaince?
L102: Respectivily to what?
Material and methods: This seems to me like an experiment aiming to evaluate animals performance troughout the levels of Ca offered than an requeirements experiment. You determined the Ca retention only based on the intake minus excreted. Where is the determination of Ca in the bone, muscle and others corporal tecides? How did you created the equations used to determine the Ca requeirements?
L147 and Table 1: IBW didn't show difference. Ok, this was expected? I didn't get why evaluate the IBW if there is no treatment applied. IBW generaly is used as covariable.
L209-232: Ok, but why? You didn't explained why the greater level of Ca can or can't influence DM intake. Is because cause physiological problems? Is because decrease palatability of the diet?
L233-252: But how? Why? The level can impact animals physiology? The metabolic regulations methods?
Discussion: You're disscussing a lot about the ratio Ca:P, however, did you measured that? In material and methods you didn't insert this information, text or table. There is a lack of discussion explaining properly the results.
Reviewer 2 Report
Comments and Suggestions for Authors
It i concluded that: integrating growth performance data, calcium utilization efficiency, and serum biochemical indexes, the calcium requirement of growing Yunnan semi-fine wool rams can be accurately determined and implemented in their dietary management. But, it would be interesting to provide data on the calcium levels in the diets that Yunnan sheep typically receive under normal production conditions. Are the systems extensive or controlled indoors?
This information would highlight the importance of the work done, because if the calcium levels in commercial breeding conditions are adequate, the supplementation recommended by the authors would not be justified.
In the end, the reasons for the great variability in calcium levels detected in the study are not clear. Is it a problem with the experimental methodology, or are there significant genetic differences between the experimental groups?
Reviewer 3 Report
Comments and Suggestions for Authors
Thank you for your effort in conducting the study on the calcium requirement of growing Yunnan semi-fine wool rams (Ovis aries) based on growth performance, calcium utilization, and serum biochemical indexes. The findings hold potential significance for the scientific community. However, there are certain areas that should be addressed to enhance the manuscript's potential for publication.
1. The background of the study appears to be somewhat lacking in depth. Strengthening this section will improve the context and relevance of your research.
2. In several places, including the abstract, the term “g/kg BW0.75” is used. The meaning of this should be clarified for readers.
3. The conclusions section needs improvement to provide a clearer explanation of the study’s results, significance, limitations, and implications for future research.
Addressing these points will likely enhance the manuscript’s quality and increase its chances of acceptance.
Round 2
Reviewer 1 Report
Comments and Suggestions for Authors
Well done.